# Effects of the Interaction between Dietary Vitamin D_3_ and Vitamin K_3_ on Growth, Skeletal Anomalies, and Expression of Bone and Calcium Metabolism-Related Genes in Juvenile Gilthead Seabream (*Sparus aurata*)

**DOI:** 10.3390/ani14192808

**Published:** 2024-09-28

**Authors:** Ulaganathan Sivagurunathan, Marisol Izquierdo, Yiyen Tseng, Philip Antony Jesu Prabhu, María Jesús Zamorano, Lidia Robaina, David Domínguez

**Affiliations:** 1Grupo de Investigación en Acuicultura (GIA), Instituto Universitario en Acuicultura Sostenible y Ecosistemas Marinos (IU-ECOAQUA), University of Las Palmas de Gran Canaria, Carretera de Taliarte, s/n, 35200 Telde, Spain; marisol.izquierdo@ulpgc.es (M.I.); tyiyen@gmail.com (Y.T.); mariajesus.zamorano@ulpgc.es (M.J.Z.); lidia.robaina@ulpgc.es (L.R.); david.dominguez@ulpgc.es (D.D.); 2Institute of Marine Research (IMR), Fish Nutrition Program, 5005 Bergen, Norway; antony.philip@nofima.no; 3Nutrition and Feed Technology Group, Nofima, 5141 Bergen, Norway

**Keywords:** vitamin D_3_, vitamin K_3_, gilthead seabream, bone biomarker, skeletal anomalies, calcium regulating genes

## Abstract

**Simple Summary:**

Vitamin D_3_ and vitamin K_3_ each play a crucial role in the growth, skeletal development, and regulation of bone biomarkers and calcium homeostasis in larval and juvenile gilthead seabream. Although their interaction has been shown to influence these parameters in animals and humans, there is limited research on this interaction in fish. In this study, juvenile gilthead seabream was fed diets with varying combinations of vitamin D_3_ and K_3_. The results showed no significant effects on growth, serum calcitriol levels, or morphometric parameters. However, a significant impact was observed on bone biomarkers and calcium-regulating genes across different tissues. Additionally, there was an increasing tendency of skeletal anomalies with higher vitamin levels. These findings suggest that, while dietary vitamin D_3_ and K_3_ can modulate bone biomarkers and calcium-regulating genes in fish, they do not significantly influence growth or serum calcitriol, likely due to the size and developmental stage of the fish. Based on this, we recommend considering vitamin D_3_ and K_3_ in diets to support skeletal health but note that they may not yield substantial changes in growth outcomes for juvenile gilthead seabream.

**Abstract:**

The interaction between vitamin D and vitamin K is crucial for regulating bone metabolism and maintaining calcium homeostasis across diverse animal species due to their complementary roles in calcium metabolism and bone health. However, research on this interaction of vitamin D and K in fish, particularly Mediterranean species like gilthead seabream, is limited or not studied. This study aimed to understand the effects of different dietary combinations of vitamin D_3_ and K_3_ on juvenile gilthead seabream. Accordingly, seabream juveniles were fed with varying combinations of vitamin D_3_/vitamin K_3_ (mg/kg diet) for 3 months: (0.07/0.01), (0.20/0.58), (0.19/1.65), (0.51/0.74), (0.56/1.00). At the end of the trial, survival, growth, body morphology, serum calcitriol, and vertebral mineral composition remained unaffected by varying vitamin levels, while gene expression patterns related to bone formation, resorption, and calcium regulation in various tissues were significantly influenced by both vitamins and their interaction. Gilthead seabream juveniles fed the 0.07/0.01 mg/kg diet upregulated calcium-regulating genes in the gills, indicating an effort to enhance calcium absorption to compensate for dietary deficiencies. Conversely, an increase in vitamin D_3_ and K_3_ up to 0.19 and 1.65 mg/kg, respectively, upregulated bone formation, bone remodeling, and calcium homeostasis-related gene expression in vertebra and other tissues. On the contrary, a dietary increase in these vitamins up to 0.56 mg/kg vitamin D_3_ and 1.00 mg/kg vitamin K_3_ downregulated calcium metabolism-related genes in tissues, suggesting an adverse interaction resulting from elevated levels of these vitamins in the diet. Hence, sustaining an equilibrium in the dietary intake of vitamin D_3_ and vitamin K_3_, in an appropriately combined form, may potentially induce interactions between the vitamins, contributing to favorable effects on bone development and calcium regulation in gilthead seabream juveniles.

## 1. Introduction

Vitamin D and K are fat-soluble vitamins that play a crucial role in bone development, cardiovascular health, and calcium homeostasis. In their active form, these vitamins are synergistic with each other and involved in numerous metabolic and physiological functions in animals and humans [1,2]. Vitamin D is an essential nutrient for fish since they are not able to synthesize it [3], except for a study in laboratory-reared rainbow trout (*Oncorhynchus mykiss*) [4]. Vitamin D ingested by the fish is primarily stored in the liver, more so than in other tissues, predominantly in the form of vitamin D_3_ [3], which makes fish a rich source of vitamin D. The vitamin D_3_ in fish gets hydroxylated in the liver [5,6] and kidney [7] using specific enzymes that result in calcitriol production. Calcitriol, the active metabolite of vitamin D_3_, is conveyed by the vitamin D binding protein to carry out its function via the vitamin D receptor (VDR) in target tissues [8,9,10]. Calcitriol binds with high affinity to VDR and activates the VDR to carry out tissue-specific classical and non-classical effects in fish species such as calcium homeostasis, bone health, and immune system regulation [11,12,13]. In addition to its receptor function, VDR also helps in the feedback mechanism to maintain vitamin D activity, suggesting a functional divergence of the receptor such as cell growth and differentiation, muscle function, cardiovascular health, and neurological function [14,15,16,17]. One of the well-known classical actions of vitamin D is to maintain calcium and phosphate homeostasis in fish [18,19]. Vitamin D_3_ actively participates in maintaining the plasma calcium level through intestinal, kidney, and bone calcium absorption and by other endocrine factors such as stanniocalcin and parathyroid hormone-related protein in fish [20,21,22,23]. For example, increasing vitamin D_3_ causes hypercalcemia in common carp (*Cyprinus carpio*) [24], rohu (*Labeo rohita*) [18], Mozambique tilapia (*Oreochromis mossambicus*) [25], gilthead seabream larvae (*Sparus aurata* [26]), and Atlantic cod (*Gadus morhua* [27]). On the contrary, a recent study showed no changes in serum calcium level in turbot (*Scophthalmus maximus*) fed increased vitamin D_3_ levels [28]. Additionally, vitamin D_3_ increases plasma phosphate in American eel (*Anguilla rostrata*), common carp (*Cyprinus* carp), and Japanese sea bass (*Lateo labrax* japonicas) [3], as well as whole-body phosphorus in *Sparus aurata* larvae [26]. However, no changes were reported in Mozambique tilapia [29] and rainbow trout (*Oncorhynchus mykiss* [30]). In addition, vitamin D functions in various tissues for the absorption of calcium and phosphate to maintain homeostasis in fish. In the intestine, vitamin D stimulates the calcium influx and increases the intestinal calcium uptake in fish species [31,32,33]. In addition, vitamin D metabolites are involved in calcium influx with the external calcium concentration through the branchial epithelial calcium channel and other endocrine factors in fish [20,34,35,36]. In fish, vitamin D metabolites play multifaceted roles. They are implicated in lipase activity and phosphate reabsorption in the kidney [37]. Moreover, when administered through either an optimal dietary content or intraperitoneal injection, vitamin D_3_ induces an elevation in the rate of bone mineralization in fish larvae [26,38]. In juveniles, vitamin D_3_ takes part in the process of new bone formation by activating osteoblast and osteoclast cells [13,39,40,41]. Apart from vitamin D, other endocrine factors like stanniocalcin and parathyroid hormone-like proteins are also involved in bone calcium absorption and resorption [18,42,43]. Vitamin D requirement in fish varies with species and developmental stage. In gilthead seabream larvae, the requirement of vitamin D_3_ is around 0.025–0.030 mg/kg [26], while gilthead seabream juveniles require 0.29 mg/kg [41]. The vitamin D_3_ requirements in other species have been briefly evaluated in various studies. [3,26,44]. Vitamin D deficiency in fish causes negative effects on growth, reduces bone mineralization, increases skeletal anomalies, hypocalcemia, and reduces survival [38,45,46]. Meanwhile, excess vitamin D cause hypercalcemia and hyperphosphatemia and increase bone mineralization, skeletal anomalies, and mortality in fish species [8,26,38,47]. Hence, an optimum level of vitamin D would maintain calcium homeostasis and normal skeletal development in fish.

Vitamin K is another lipid-soluble vitamin known for its function in blood coagulation, cardiovascular health, neurodegenerative diseases, and activation of vitamin K-dependent proteins (VKDP), bone and muscle metabolism, and calcium homeostasis [48,49,50,51,52]. Vitamin K is available to fish in three different forms: vitamin K1 (phylloquinone, plant origin), vitamin K2 (menaquinones, microbial origin), and vitamin K_3_ (menadione, synthetic origin) [53]. In commercial feeds, vitamin K_3_ is the commonly used dietary source of vitamin K. The ingested vitamin K_3_ is alkylated enzymatically to MK-4 in tissues to perform various metabolic functions in the body [54,55]. In fish, vitamin D promotes the production of VKDP, which are bone Gla-protein (BGP/osteocalcin) and matrix Gla-protein (MGP), that undergo posttranslational modification in the presence of vitamin K for its activation [2,56]. The activated BGP is involved in bone mineralization and MGP helps in the inhibition of soft tissue calcification [5,57]. The expression of VKDPs is found to be present in vertebrae, fin, gill, heart, intestine, and other vital tissues of fish, to regulate mineralization and calcium homeostasis [5,58,59,60]. In fish, vitamin K deficiency has a severe impact on bone, causing abnormal bone development, higher incidence of skeletal anomalies, reduced mineralization, and increased mortality [5,59,61,62,63,64]. Meanwhile, increasing vitamin K_3_ in the diet caused toxicity in mammals [65,66], while in fish species, the reports are contradictory and no negative effects on growth, mortality, and bone health have been observed [55,59]. However, in gilthead seabream larvae fed high vitamin K_3_ levels (58.51 mg/kg), survival and vertebral mineralization are reduced [67] and in gilthead seabream juveniles, 23 mg/kg growth is negatively affected [5]. Therefore, vitamin K requirements in fish species vary with species and developmental stage. The requirements of vitamin K in fish have been briefly reviewed in several studies [44,50].

The potential interaction between these two vitamins has only recently caught the attention of researchers aiming to understand the synergism between them in improving bone and cardiovascular health and regulating calcium homeostasis in humans [68,69]. The combined effect of vitamin D and vitamin K suggests that vitamin D promotes VKDP synthesis, and vitamin K helps in the posttranslational modification of these proteins to induce bone formation and limit vascular calcification in humans and mice [1,11]. The optimum supplementation of vitamin D along with vitamin K stimulated intestinal calcium absorption, prevented kidney disease, reduced fracture, and prevented bone loss in humans [70,71,72]. Within the bone, the osteoblast was identified to possess enzymes involved in the vitamin K cycle, specifically γ-glutamyl carboxylase (GGCX) and vitamin K epoxide reductase (VKORC1). These enzymes play a role in the aggregation and mineralization of osteocalcin in bones, working in conjunction with the activity of vitamin D_3_ [2,73]. Similarly, the investigation of vascular function and calcification in humans indicates a positive effect of vitamins D and K on cardiovascular health [2,11,74]. However, in human studies, vitamin K was supplemented in the form of vitamin K2 along with vitamin D_3_. In fish, studies on vitamin D and K interaction are scarce or limited. In Atlantic salmon (*Salmo salar*), a combination of vitamin D_3_, vitamin K_3_, calcium, and environmental factors did not affect growth or bone mineralization [63]. Recently, the interaction of dietary vitamin D_3_ and vitamin K_3_ was also studied on gilthead seabream larvae, showing that an increase in both vitamins reduces larval growth and survival and causes abnormal expression of bone biomarkers and calcium regulating genes [75]. This study further elucidates the significance of bone biomarkers and calcium-regulating genes in the modulation of bone formation and the maintenance of calcium homeostasis in fish. The focus of interaction studies has primarily been on gilthead seabream larvae, leaving a gap in research for gilthead seabream juveniles. Considering the commercial importance of these fish in the Mediterranean region and Europe [76,77], where they hold significant market value and stand out in aquaculture production by replacing fish meal and fish oil in their diet, optimizing the inclusion of vitamin D_3_ and K_3_ in their diet becomes crucial. The functions and metabolism of these vitamins exhibit variations across different developmental stages of fish, emphasizing the need to customize the optimization of vitamin D_3_ and vitamin K_3_ for gilthead seabream at various growth stages. In response to this research gap, the present study aims to enhance our understanding of the interplay between dietary vitamin D_3_ and vitamin K_3_, specifically concerning growth, bone biomarkers, and genes that regulate calcium metabolism in different tissues of commercially important gilthead seabream juveniles.

## 2. Materials and Methods

### 2.1. Experimental Diets and Proximate Composition

Five plant-based diets containing low fish meal (FM—10%) and fish oil (FO—6%) were formulated, combining two levels of vitamin D_3_ (Cholecalciferol: CAS-67970, Sigma-Aldrich, Saint Louis, MO, USA) with two levels of vitamin K_3_ (Menadione: CAS-58275, Sigma-Aldrich, Saint Louis, MO, USA), and one diet without supplementation (Table 1). The diet formulation and vitamin levels in the present study were determined based on the optimal levels for vitamin D_3_ and K_3_ established in our previous laboratory study [5,41]. The diets were formulated using high-quality ingredients. Although the formulation originated from a company design intended to enhance result applicability, it is not a commercial product. The experimental diets were prepared at GIA, ECOAQUA, ULPGC, Spain. The diet preparation process encompassed the grinding of raw materials, precise weighing, homogenous mixing, extrusion, and subsequent drying. In consideration of the sensitivity of vitamins to temperature-induced alterations, the mixture underwent pelletization without steam, employing a laboratory pelleting machine (California Pellet Mill, CPM, 2HP mod 8.3, Crawfordsville, IN, USA). Following pelletization, the pellets were subjected to a drying period of 24 h at 32 °C using a dryer. Subsequently, the pellets were stored in a dark, refrigerated chamber at a temperature below 10 °C until their utilization. Prepared diets were analyzed for proximate composition (Table 1) and formulated to be isoproteic (44.52 ± 0.20%) and isolipidic (22.46 ± 0.81%), covering all known essential nutrients for gilthead seabream juvenile. Moisture, ash, and crude protein were determined according to [78], and crude lipid contents by the method of [79] (Table 1). The determination of fat-soluble vitamin contents involved two stages: preparative high-performance liquid chromatography (HPLC) and analytical HPLC. For preparative HPLC, the setup included an isocratic pump (Spectra system P 1000, Markham, ON, Canada), Gilson 112 UV Detector, (Gilson, Inc, Middleton, WI, USA), and an injector with a 200 µL loop (Rheodyne 7725, Rohnert Park, CA, USA), all controlled by the Lab data software Chromeleon 7.3.2. In the analytical HPLC phase, an isocratic pump (LaChrom: Merck HITACHI L7100, Tokyo, Japan), a UV/VIS detector (LaChrom: Merck HITACHI L7420, Tokyo, Japan), and an Autosampler (Gilson 234, Gilson, Inc., Middleton, WI, USA) were employed. The integration of data was facilitated by the Chromeleon software 7.3.2. These analyses were conducted at the Institute of Marine Research (IMR) in Bergen, Norway [80].

### 2.2. Juvenile Rearing

The fish used in this study came from PROGENSA, a selection program run at a national level in Spain for gilthead seabream for almost 10 years, coordinated by our research group. Our breeders are managed under conditions like industrial practices, including their feeding, composition, and overall management, ensuring their genetic fitness is maintained at an optimal level. Fish weighing around 40–60 g were fed with a non-supplemented diet containing 0.07 mg of vitamin D_3_/kg and 0.01 mg of vitamin K_3_/kg for a period of one month to normalize the dietary intake of vitamin D and K in the body. In the subsequent month, the fish underwent assessment for irregular body shape and any infection through visual observation, serving as an indicative marker for the presence of skeletal anomalies and disease. Those exhibiting a regular body shape, devoid of malformations and free from disease, were subsequently selected for inclusion in the experimental trial. Three hundred gilthead seabream juveniles, weighing 72.63 ± 0.33 g average body weight, were distributed into 15 tanks (170 L/tank). Fish were fed until apparent satiation three times daily, at a ratio of 2% body weight per day for three months. Water quality, temperature (21.9 ± 0.2 °C), light (12 h light—natural photoperiod), and other abiotic factors were regularly checked and maintained to be constant.

### 2.3. Sample Collection and Analyses

Gilthead seabream juveniles were fasted for 24 h before sampling and sacrificed by adding a high dose of clove oil (Guinama S.L.U., Valencia, Spain). All fish were obtained to study growth performance and survival rate. Among them, blood was collected from three fish per tank for serum calcitriol analysis, three fish for vertebral mineral composition, ten fish per tank for X-ray to identify the occurrence of skeletal anomalies, and tissue samples from three fish per tank for gene expression studies.

### 2.4. Growth Performance

Fish were sampled every two weeks to study their growth performance by measuring the total length (cm) and body weight (g) of all the fish from each tank. At the end of the experimental period, fish were sampled for survival rate, final total length, and body weight, to calculate Weight Gain (WG), Specific Growth Rate (SGR), Feed Conversion Ratio (FCR), Feed Intake (FI), Protein Efficiency Ratio (PER), and Condition Factor (CF).
$$\mathrm{Survival}\;\mathrm{rate}=\left(\frac{\mathrm{Final}\;\mathrm{number}\;\mathrm{of}\;\mathrm{fish}}{\mathrm{Initial}\;\mathrm{number}\;\mathrm{of}\;\mathrm{fish}}\right)\times 100$$
$$\mathrm{WG}\;(\%)=100\times \left(\frac{\mathrm{final}\;\mathrm{weight}-\mathrm{initial}\;\mathrm{weight}}{\mathrm{initial}\;\mathrm{weight}}\right)$$
$$\mathrm{SGR}\;(\%)=\frac{(\mathrm{Ln}W_{2}-\mathrm{Ln}W_{1})}{\mathrm{Experimental}\;\mathrm{days}}\times 100\;(\mathrm{where},\;W_{2}\;=\;\mathrm{Final}\;\mathrm{weight};\;W_{1}\;=\;\mathrm{Initial}\;\mathrm{weight})$$
$$\mathrm{FCR}=\frac{\mathrm{Feed}\;\mathrm{intake}\;}{\mathrm{Fish}\;\mathrm{weight}\;\mathrm{gained}\;}$$
$$\mathrm{FI}=\frac{\mathrm{Total}\;\mathrm{feed}\;\mathrm{given}\;(g)\;}{Total\;number\;of\;fish}$$
$$\mathrm{PER}=\frac{\mathrm{Fish}\;\mathrm{weight}\;\mathrm{gained}}{\mathrm{protein}\;\mathrm{fed}}\;\mathrm{where},\;\mathrm{protein}\;\mathrm{fed}=\frac{\mathrm{Total}\;\mathrm{feed}\;\mathrm{consumed}\;\times \mathrm{Crude}\;\mathrm{protein}\;\mathrm{in}\;\mathrm{feed}}{100}$$
$$\mathrm{CF}=\left(\frac{\mathrm{Final}\;\mathrm{weight}}{{\mathrm{Final}\;\mathrm{length}}^{3}}\right)\times 100$$


### 2.5. Serum Calcitriol

Serum samples were obtained by collecting blood from the caudal sinus of gilthead seabream juveniles using sterile 1 mL plastic syringes and stored in Eppendorf tubes without any disturbance for 15–30 min at room temperature. Once the blood was clotted, serum was separated by centrifuging at 1000–2000 g for 10 min at 4 °C. Then, the serum was pipetted into new tubes for calcitriol analysis. To measure the serum calcitriol, a competitive enzyme immunoassay was performed using the BIOMATIK DHVD3 ELISA kit (Biomatik corporation, Wilmington, DE, USA). Sample collection, reagent preparation, sample preparation, and assay procedure were conducted according to the manufacturer’s instructions.

### 2.6. Vertebral Mineral Composition

The whole vertebral column was collected from gilthead seabream juveniles and cleaned thoroughly until the attached muscle and fat tissues were wiped off. The cleaned vertebrae were lyophilized using a freeze dryer to determine the calcium and phosphorus content. The samples were analyzed at IMR, Bergen, Norway. The process of mineral determination followed acidification and digestion of the samples (Microwave digester; MarsXpress, CEM, Kamp-Lintfort, Germany), and was determined using Inductively Coupled Plasma Mass Spectrometry (ICP-MS) with an auto sampler (FAST SC-4Q DX, Waltham, MA, USA) [81].

### 2.7. Skeletal Anomalies

To determine the skeletal development of the sampled juvenile fish, whole bodies were vacuum-packed for radiography analysis. Bone development and skeletal anomalies in juveniles were determined by performing an X-ray using the GE Senographe DMR Plus mammography system (Chicago, IL, USA). The X-ray frequency was adjusted to 32 kV and 45.0 mAs to obtain clear images of the fish samples using the SharpIQ grid system (Chicago, IL, USA). The radiographed images were processed for better visualization and evaluation using Codonics Clarity Viewer 7.5 SP 6, (Codonics, OH, USA). The processed images were evaluated for different severe skeletal anomalies and their classification based on [82].

### 2.8. Morphometric Analysis

To determine the morphometric traits of gilthead seabream juveniles among the different treatments, 10–12 fish per tank were collected and photographed in a dark room using an Olympus digital camera (FE230/X790, Olympus lens 6.3 to 18.9 mm, f3.1 to 5.9, equivalent to 38 to 114 mm on a 35 mm camera, Olympus corporation, Shinjuku, Tokyo, Japan) that was fixed to the table. The camera was mounted at a specific fixed distance from the object with a fixed aperture (3X optical zoom lens, f3.1 at wide angle) to obtain homogenous images throughout this study. Two tubular fluorescent lamps (16 mm H14 W/827 with 2700 K color temperature) were placed beside the object at a specific distance to illuminate the object evenly. This reduced the shadow error and improved the accuracy. Meanwhile, the table was covered with red background sheets for lateral images. Before photographing the fish, the image was calibrated with three photos of a black square sheet (20 cm × 20 cm) and then continued with the fish samples along with their labels. The captured images were processed by IMAFISH_ML, a fully automated image analysis software using MATLAB^®^ v.7.5, Natick, MA, USA. The morphometric traits were analyzed according to [83].

### 2.9. Gene Expression

Total RNA was obtained from 60–80 mg of tissue samples such as liver, intestine, vertebrae, gill, heart, and kidney, respectively, using TRI Reagent (Sigma-Aldrich, Sant Louis, MO, USA) and purified using the RNeasy Mini extraction kit (Qiagen, Hilden, Germany). Tissue samples were mixed with TRI Reagent and an autoclaved metal ball was added to it for homogenization. TissueLyzer-II (Qiagen, Germany) was used to homogenize the tissue. The lysed samples were centrifuged at 13,000× *g* for 1 min and the liquid phase of TRI-reagent was separated into new Eppendorf tubes. To this reagent, 200 µL of chloroform was added, vortexed, and centrifuged at 12,000× *g* for 15 min, at 4 °C. The upper aqueous phase containing RNA was collected into a new sterile or autoclaved Eppendorf tube and 75% ethanol was mixed into it for RNA extraction. The extracted RNA phase was transferred into an RNeasy spin column followed by RW1 and RPE (Qiagen, Germany) buffer washes, according to manufacturer protocol. The filtered RNA was eluted from the RNeasy spin column using nuclease-free water. The eluted RNA was examined for quality and quantity using a NanoDrop 1000 Spectrophotometer (Thermo Fisher Scientific, Wilmington, NC, USA). From the eluted RNA, complementary DNA (cDNA) was synthesized using the iScript cDNA Synthesis Kit (Bio-Rad, Hercules, CA, USA) in an iCycler thermal cycler (Bio-Rad, USA) following the manufacturer’s instructions. The synthesized cDNA was diluted to conduct RT-PCR using different primers designed for specific genes of interest. The primers of housekeeping genes such as *β-actin2* (Beta actin 2), *ef1α* (Elongation factor 1 alpha), *rpl27* (Ribosomal Protein L27), and target genes in this study were listed in Table 2. The housekeeping genes for each specific tissue were selected through a comparative analysis using BestKeeper version 1 (https://www.gene-quantification.de/bestkeeper.html, accessed on 4 March 2024) and RefFinder software (https://www.ciidirsinaloa.com.mx/RefFinder-master/, accessed on 4 March 2024), with β-actin2 ultimately chosen as it demonstrated minimal significant differences across all tissues. The gene expression study was determined by RT-PCR in an iQ5 Multicolor Real-Time PCR detection system (Bio-Rad, USA). The RT-PCR conditions for the present study were as follows: 95 °C for 3 min and 30 s followed by 40 cycles of 95 °C for 15 s, annealing temperature for 30 s as mentioned in Table 2 for each gene, and 72 °C for 30 s; 95 °C for 1 min, and a final denaturation step from 58 to 95 °C for 10 s.

### 2.10. Statistical Analysis

All data were interpreted as means ± S.D. using IBM SPSS Statistics v26.0. (IBM Corp., Chicago, IL, USA). Levene’s statistic was performed to test for normality and homogeneity of variance among the calculated data. If the samples did not test for Levene’s statistic, they were analyzed using non-paramatric tests to determine the significant differences among the groups. Accordingly, proximate composition, growth performance, serum calcitriol, frequency of skeletal anomalies, and relative gene expression were analyzed using one-way ANOVA, to determine the dietary effect among the groups by comparing the means and to provide statistical differences using Tukey’s posthoc test (*p* < 0.05). Similarly, a two-way ANOVA was performed to understand the individual dietary effect and the interaction effect of different vitamin levels in different dietary groups (*p* < 0.05). Skeletal anomalies were analyzed using non-parametric Kruskal-Walli’s test. The Livak method (2^−ΔΔct^) was performed to normalize the relative gene expression of tissue samples. The magnitude of up and down-regulation of target genes was compared by conducting a heatmap differentiation using GraphPad Prism 9 (GraphPad software, San Diego, CA, USA) and their hierarchical clustering of the different biological replicates and experimental groups was presented using PAST 4.03 software (LO4D.com, London, UK).

## 3. Results

### 3.1. Experimental Diets and Proximate Composition

The formulated diets showed no significant differences in their proximate composition (Table 1) and fulfilled the protein, lipid, and energy requirements of gilthead seabream juveniles. The diets provided to juvenile gilthead seabream were well-received from the start of the trial, resulting in minimal or no leftover feed accumulating at the bottom of the tanks. The feeding ratio and feeding regime were adjusted according to the daily feed intake.

### 3.2. Growth Performance

During the juvenile rearing period, no mortalities were observed. After 3 months of feeding, there were no significant differences in growth performance, including final body weight (FBW), weight gain (WG), specific growth rate (SGR), feed conversion ratio (FCR), feed intake (FI), protein efficiency ratio (PER), and condition factor (CF) of gilthead seabream juveniles (Table 3). Similarly, no dietary interaction effect was revealed by the two-way ANOVA analysis (Table 3).

### 3.3. Serum Calcitriol

Serum calcitriol levels did not significantly (*p* > 0.05) differ among juveniles fed the different vitamin D_3_ and vitamin K_3_ levels (Table 4). The two-way ANOVA showed no interaction of vitamin D_3_ and vitamin K_3_ on serum calcitriol (Table 4).

### 3.4. Vertebral Mineral Composition

Vertebral calcium and phosphorus composition in gilthead seabream juveniles were not altered by dietary vitamin D_3_ and vitamin K_3_ or their interaction (Table 4).

### 3.5. Skeletal Anomalies

Figure 1 illustrates the occurrence of various skeletal anomalies in gilthead seabream juveniles, providing a visual representation of the observed anomalies. There were no significant differences (*p* > 0.05) in the frequency of the diverse skeletal anomalies studied among fish that were fed different diets (Table 5). Nevertheless, fish that were fed diet 0.19/1.65 showed a tendency to a lower incidence of severe anomalies (2.77%), whereas fish that were fed the 0.56/1.00 diet showed a tendency for a higher incidence (25.00%, Table 5, Figure 2).

### 3.6. Morphometric Analysis

Gilthead seabream juveniles fed with different combinations of dietary vitamin D_3_ and K_3_ presented no significant differences in the morphometric traits of the fish. Fish body shape, fillet area percentage, and other morphometric traits were not significantly different among the dietary groups as denoted by two-way ANOVA analysis (Table 6).

### 3.7. Gene Expression

At the end of the experimental period, the highest values of bone morphogenic protein 2 (*bmp2*) and osterix (*osx*) were found in fish that were fed diet 0.19/1.65, being significantly (*p* < 0.05) higher than those in fish that were fed diet 0.20/0.58 and, for *osx*, diet 0.07/0.01 (Table 7). Additionally, for both biomarkers, two-way ANOVA denoted a significant (*p* < 0.05) up-regulatory effect of vitamin K_3_ and a significant (*p* < 0.05) interaction between both vitamins. The expression of the alkaline phosphatase gene (*alp*) was significantly (*p* < 0.05) upregulated by the elevation of dietary vitamin D_3_ and vitamin K_3_ as denoted by two-way ANOVA (Table 7). Moreover, the increase in vitamin K_3_ at the highest vitamin D_3_ level downregulated *alp*, reflecting the interaction (*p* < 0.05) between these vitamins as shown by two-way ANOVA. Therefore, the highest *alp* expression was found in fish that were fed the 0.51/0.74 diet. On the contrary, *runx2*, osteonectin, and osteocalcin gene expressions in vertebrae were not affected by the dietary vitamin levels. An interaction (*p* < 0.05) between these vitamins was also recorded in the osteopontin expression. The osteoclast-related matrix gla protein (*mgp*) gene expression was significantly (*p* < 0.05) higher in fish that were fed the 0.19/1.65 and 0.51/0.74 diets, being upregulated by the elevation of dietary vitamin D_3_ and vitamin K_3_ as denoted by two-way ANOVA, which also showed the significant interaction of both vitamins (Table 7). In fish that were fed these two diets, the tartrate-resistant acid phosphatase (*trap*) gene expression was also significantly (*p* < 0.05) higher, with a significant (*p* < 0.05) interaction between both vitamins. In addition, the bone resorption marker cathepsin K (*ctsk*) was significantly highest in fish that were fed the 0.51/0.74 diet and tended to be upregulated by vitamin D_3_ (*p* = 0.06) as denoted by two-way ANOVA and showed a significant (*p* < 0.05) interaction between both vitamins. On the contrary, the matrix metalloproteinase 9 (*mmp9*) gene did not significantly (*p* > 0.05) differ among dietary groups. The calcium-sensing receptor (*casr*) was significantly higher in fish that were fed diet 0.19/1.65, being significantly (*p* < 0.05) upregulated by both vitamin D_3_ and vitamin K_3_, with a strong interaction between them as shown by two-way ANOVA (Table 7). Similarly, expressions of parathyroid hormone 1 receptor (*pth1r*), parathyroid hormone-related protein (*pthrp*), and stanniocalcin 2 (*stc2*) genes were highest in fish that were fed diet 0.19/1.65, being significantly (*p* < 0.05) upregulated by vitamin K_3_, with a strong interaction between them as shown by two-way ANOVA (Table 7). Therefore, the highest expression of *casr*, *pth1r,* and *pthrp* was found in fish that were fed the 0.19/1.65 diet (Table 7). As for stanniocalcin 2 (*stc2*), only vitamin K_3_ significantly upregulated this gene’s expression, which was highest in vertebrae of fish that were fed the 0.19/1.65 diet. Overall, there was an upregulation of the genes studied in the vertebral genes for fish that were fed 0.19/1.65 mg/kg, as illustrated by the heat map figures obtained (Figure 3a).

In the intestine (Table 8), *casr* expression was significantly upregulated by dietary vitamin D_3_ as denoted by two-way ANOVA, whereas a significant interaction of both vitamins was found in the expression of this gene, as well as *pth1r*, *pthrp*, and *stc2*. The highest expressions for these four genes (*p* < 0.05) were found in fish that were fed diet 0.19/1.65 and the lowest in those fed 0.07/0.01 and 0.56/1.00 diets. Therefore, the two-way ANOVA denoted a significant (*p* < 0.05) interaction between vitamin D_3_ and K_3_ on intestinal calcium regulators. The heat map analysis showed the overall upregulation of calcium-regulated genes in fish that were fed the 0.19/1.65 diet and the downregulation in fish that were fed the lowest (diet 0.07/0.01) and the highest (diet 0.56/1.00) dietary vitamins levels (Figure 4).

In the kidney, the relative expression of *vdrβ* was significantly (*p* < 0.05) upregulated with an increase in dietary vitamin D_3_, with the highest values in fish that were fed diet 0.56/1.00 (Table 9). Meanwhile, the relative expression of *stc2* was highest in fish that were fed 0.19/1.65 and lowest in fish that were fed 0.20/0.58 diet (Table 9). The two-way ANOVA showed an up-regulatory effect of dietary vitamin K_3_ on *stc2* expression and the interaction between both vitamins (Table 9).

In the liver, the calcium-regulating genes *casr, pth1r, pthrp,* and *stc2* were not affected by the different dietary vitamin levels, and no interactions were denoted by two-way ANOVA (Table 10). The heat map comparison suggested a reduced expression of calcium-related genes in fish that were fed the 0.19/1.65 diet (Figure 5).

In the gills, the expression of calcium-related genes *casr*, *pth1r*, and *stc2* was highest (*p* < 0.05) in gilthead seabream that were fed the 0.07/0.01 diet with no vitamin D_3_ and vitamin K_3_ supplementation (Table 11). The two-way ANOVA showed a down-regulatory effect of dietary vitamin K_3_ on calcium regulators (Table 11). The heat map comparison suggested an increased expression of calcium-related genes in fish that were fed the 0.07/0.01 diet (Figure 6).

## 4. Discussion

The interplay between vitamin D and vitamin K is pivotal in regulating bone metabolism and ensuring calcium homeostasis across the animal kingdom [2]. Despite the importance of the interaction of these vitamins in different animal species, there are few studies on fish [5,25,26,41,67,75], and no studies have been reported on the interaction of vitamin D_3_ and vitamin K_3_ in commercially important Mediterranean fish species. In the present study, feeding juvenile gilthead seabream a plant-based diet fortified with varying amounts of vitamin D_3_ and vitamin K_3_ did not impact growth, body morphometric traits, or serum calcitriol levels. This finding contrasts with our previous studies, which used different levels of vitamin D_3_ and vitamin K_3_ in separate experiments and observed improved growth [5,41]. The lack of growth effects in this study may be attributed to the larger size of the fish compared to those in the previous studies, suggesting that the vitamin levels had less impact on growth and morphometric changes in larger fish. However, the expression patterns of genes involved in bone formation, bone resorption, and calcium regulation in various tissues were significantly influenced by both vitamins and their interaction, consistent with findings from previous studies on vitamin D_3_ and K_3_ [5,41]. This indicates that the effects of these vitamins are more pronounced in bone and calcium-related functions in larger fish, rather than in growth and morphometric markers.

Gilthead seabream juveniles fed the 0.19/1.65 diet showed the highest expression of *bmp2* and *osx*, which were significantly up regulated by dietary vitamin K_3_ levels, in agreement with the previous study [5]. The *osx* is a master regulator of late osteogenesis [84,85,86], whereas *bmp2* is an osteochondrogenic factor that promotes bone formation and bone healing and can induce the expression of osteogenic transcription factors, such as *osx* [87,88,89]. Indeed, a significant lineal regression (R^2^ = 0.93; *p* = 0.01; Figure 3b) was found between *bmp2* and *osx* expression in the present study, both contributing to bone turnover by increasing osteoblast differentiation into mature osteoblasts [90] and upregulating the expression of their vitamin D receptor gene [91]. In addition, fish that were fed the 0.19/1.65 diet also showed a significant upregulation of the osteoclast markers, particularly mgp and trap, denoting a bone remodeling process. On one hand, *trap* expressed by osteoclasts induces bone resorption, and on the other hand, the matrix Gla protein promotes bone formation by the proliferation of osteoblasts. Hence, the intricate interplay between *trap* and *mgp* exemplifies the dynamic equilibrium in bone remodeling. While trap expressed by osteoclasts facilitates bone resorption, *mgp*, a vitamin K-dependent protein, promotes bone formation by influencing the activities of osteoblasts and ensuring the appropriate mineralization of bone tissue [92]. Moreover, the expressions of genes related to calcium metabolism such as *casr*, *pth1r, pthrp,* and *stc2* were also upregulated in fish that were fed the 0.19/1.65 diet, following significant relations among them, denoting a well-balanced cascade in the expression of these genes, required to maintain bone health [93,94,95]. The *pth1r* expression in seabream vertebrae denotes the expression of *pthrp* in osteoblasts, which activates osteoclasts for net bone turnover, as shown both in mammals [96,97] and fish [98,99,100]. Meanwhile, *casr* regulates the expression of *stc2* [101] and improves osteoblast differentiation during bone turnover [102,103], whereas *stc2* contributes to osteoblast differentiation along with bone differentiation markers [104]. Hence, both vitamin D and K are involved in the dynamic process of bone remodeling [105,106]; an imbalance in these vitamins could affect bone quality, increase the risk of fracture [1,107], and cause skeletal anomalies in fish [5,41,75]. Hence, adequate levels of these vitamins are required to maintain normal bone formation and mineralization [108,109]. Therefore, fish that were fed the 0.19/1.65 diet showed an upregulation of genes related to bone formation, bone remodeling, and calcium metabolism in a balanced relation, suggesting normal bone development, bone turnover, and calcium homeostasis in fish bone.

On the contrary, further elevation of vitamin D_3_ up to 0.56 mg/kg at a dietary vitamin K_3_ level of 1 mg/kg (0.56/1 diet) tended to increase the incidence of severe anomalies (25% severe anomalies) in comparison to fish that were fed the 0.19/1.65 diet (2.77% severe anomalies). Despite there being no significant difference, a tendency was observed with skeletal anomalies. These results are consistent with the increased skeletal anomalies in juveniles and larvae of gilthead seabream fed high levels of vitamin D_3_ [26,41]. Nevertheless, in the present study, an increase in dietary vitamin D_3_ and vitamin K_3_ significantly upregulated *alp*, the gene that codifies for the ectoenzyme in the osteoblast that regulates bone mineralization, in agreement with our previous studies in larvae [26]. High *alp* expression has been also associated with bone anomalies and healing fractures [19,89,110] because of increased bone remodeling. Therefore, in the present study, the upregulation of *alp* together with the maintenance of Ca and P levels in vertebrae could be the consequence of a bone resorption process. Nevertheless, dietary vitamin K_3_ increases up to 1 mg/kg in diets containing the highest vitamin D_3_ level (0.56/1 diet) significantly down-regulated *alp* and *trap*, denoting the inhibition of bone resorption and remodeling. Moreover, two of the calcium metabolism genes, *pthrp,* and *stc2*, were also downregulated and kept an unbalanced proportion between their expressions in vertebrae (pthrp/stc2 = 0.5), in comparison with all the other fish groups (*pthrp/stc2* = 1). These results concur with the tendency to a higher incidence of severe skeletal anomalies found in fish that were fed the 0.56/1 diet (25% severe anomalies), in comparison to fish that were fed the 0.19/1.65 diet (2.77% severe anomalies), suggesting an imbalanced expression of bone and calcium metabolism biomarkers, which portrays the negative effect of increasing these vitamins on bone health [26,38]. These results agree with the negative effect of the combined increase of vitamin D_3_ and K_3_ in gilthead seabream larvae, whose interaction caused an increased incidence of skeletal anomalies [75], despite no significant difference being observed.

Genes implicated in bone calcium metabolism and biomarkers found in vertebrae, along with genes associated with calcium regulation in the intestine, are interconnected due to their roles in preserving calcium balance and skeletal health [111,112]. Disruptions in either pathway can affect the other, potentially resulting in anomalies in bone mineralization and density [113]. Fish that were fed the 0.19/1.65 diet exhibited upregulation of calcium metabolism-related genes, suggesting efficient calcium absorption, which directly influences the availability of calcium for bone mineralization and remodeling [114]. This finding concurs with the heightened expression of calcium-related genes and bone biomarkers observed in fish that were fed the 0.19/1.65 diet, contributing to the mitigation of skeletal anomalies (2.77%) [115]. Intestinal *casr* regulates calcium balance through bicarbonate secretion and the enhancement of endocrine factors such as *pthrp* and *stc2* in the intestine [116,117], consistent with the significant associations noted among the expression of these genes in the present study. Conversely, calcium metabolism-related genes were downregulated in fish that were fed the 0.56/1.00 diet, indicating an adverse interaction resulting from elevated levels of these vitamins, which likely influenced the heightened tendency of skeletal anomalies (25%) in this group. In the kidney, fish that were fed a high dietary vitamin D_3_ showed an upregulation in the gene responsible for the vitamin D receptor, *vdrβ*. This receptor serves as the binding site for the active form of vitamin D, regulating the expression of genes involved in calcium homeostasis [118]. Furthermore, in fish that were fed the 0.19/1.65 diet, there was an upregulation in the expression of *stc2*, indicating enhanced calcium reabsorption in the kidney to maintain blood calcium levels [119,120]. Given the kidney’s involvement in calcium handling through genes like *vdrβ* and *stc2*, which are tightly regulated by systemic factors and feedback mechanisms, both genes play a crucial role in maintaining calcium homeostasis in fish [121]. Additionally, studies suggest that the kidneys interact with other organs involved in calcium homeostasis, such as the bones and intestines, to ensure proper regulation of calcium balance throughout the body [122]. However, the upregulated expression of *vdrβ* and *stc2* in the kidney did not affect serum calcitriol and vertebral calcium levels, suggesting local synthesis of calcitriol without causing significant changes in serum calcitriol levels due to feedback regulation mechanisms and local utilization of calcitriol in the kidney [3,123]. Furthermore, the calcium-related genes in the liver remained unaltered across all groups fed varying levels of dietary vitamin D_3_ and vitamin K_3_, implying the existence of compensatory mechanisms or feedback loops within the regulatory network of calcium homeostasis [28,124]. It is plausible that other tissues, such as the kidney and intestine, may compensate for any changes in liver gene expression to uphold overall calcium balance. The findings of this study are consistent with the idea that the liver’s main function is not directly linked to calcium metabolism [125,126]. This is supported by the absence of changes in calcium-related gene expression observed in the liver of gilthead seabream juveniles.

Being a marine fish species, gilthead seabream likely receives sufficient calcium from the external environment through the gills and diet to maintain calcium homeostasis [94,95]. In the gills, fish that were fed a diet without supplementation of vitamin D_3_ or K_3_ (0.07/0.01) exhibited an upregulation of *pth1r*, *casr*, and *stc2* genes, suggesting a compensatory response to regulate calcium uptake. This compensatory effect is consistent with previous findings in other species suggesting dietary deficiencies [93,98,114]. Despite reports of serum hypercalcemia and hyperphosphatemia in freshwater fish administered with calcitriol [127,128], such effects were not observed in the present study. This suggests that increasing dietary vitamin D_3_ up to 0.56 mg/kg and vitamin K_3_ up to 1.00 mg/kg did not alter calcium homeostasis in gilthead seabream. Additionally, the gills may possess mechanisms for the local regulation of calcium levels independent of systemic calcium levels, which could explain the increase in calcium-related genes in fish that were fed the 0.07/0.01 diet. However, this upregulation in fish that were fed the 0.07/0.01 diet could potentially impact skeletal development and tended to increase total skeletal anomalies (20.27%) in this group.

The morphological quality of gilthead seabream in the present study was not affected by the occurrence of skeletal anomalies when fed with different levels of vitamin D_3_ and vitamin K_3_. This suggests that these vitamins, at the dietary levels tested in the present feeding trial, may not have negative effects on morphometric traits, which would help commercial farms to select and harvest the fish in a cost-effective manner [129]. In agreement, neither growth nor survival was affected by feeding seabream juveniles with 0.07–0.56 mg/kg vitamin D_3_ in combination with 0.01–1.65 mg/kg vitamin K_3_, despite almost tripling their initial weight. These results suggest that even the basal supply of vitamins was sufficient to maintain the survival and growth of juvenile gilthead seabream. This is consistent with the lack of effect of these vitamins on the survival and growth of 20 g juvenile gilthead seabream fed separately with similar levels of dietary vitamin D_3_ (0.1–0.65 mg/kg) [41] and vitamin K_3_ (0.1–3 mg/kg) [5] in plant-based diets. Similarly, in larvae of gilthead seabream, an increase in either dietary vitamin D_3_ or vitamin K_3_ over the dietary levels considered in the present study negatively affected larval survival or growth [26,67,75]. Therefore, fish survival and growth may be affected by dietary vitamin D_3_ [3,8,13,19,26,38,45,47,110,130] and dietary vitamin K_3_ [5,50,60,131,132,133], depending on the species, the developmental stage, and the dietary vitamins levels studied. In addition, in the present study, neither the interaction of these vitamins affected the growth or survival of juvenile gilthead seabream. On the contrary, in larvae of gilthead seabream, feeding rising vitamin D_3_ in conjunction with increased vitamin K_3_ resulted in reduced survival and larval growth [75]. These results suggested a higher sensitivity of younger seabream stages to the interaction of vitamin D_3_ and vitamin K_3_ than at the juvenile stage.

## 5. Conclusions

In summary, the present study showed that the survival, growth, morphological quality, serum calcitriol, and vertebral mineral composition of gilthead seabream were not significantly affected by feeding juveniles with varying combinations of dietary vitamin D_3_ and K_3_ within the ranges of 0.07–0.56 mg/kg and 0.01–1.65 mg/kg, respectively. However, a combined increase in vitamins D_3_ and K_3_ up to 0.19 and 1.65 mg/kg significantly upregulated genes related to calcium metabolism and bone formation, indicating a positive interaction between vitamin D_3_ and K_3_ on bone health and calcium metabolism. Conversely, diets with low vitamin D_3_ and K_3_ levels, up to 0.07 mg/kg and 0.01 mg/kg, downregulated calcium metabolism genes in vertebrae and intestine, while showing an upregulation in gills for calcium uptake as compensatory intake. Additionally, excessive elevation of vitamin D_3_ and K_3_, up to 0.56 mg/kg and 1 mg/kg in diet, downregulated calcium metabolism-related genes in the intestine, gill, and bone, suggesting an adverse interaction resulting from elevated levels of these vitamins in the diet. Therefore, the present study underscores the importance of maintaining a balanced intake of both vitamin D_3_ and vitamin K_3_ in the diet, as an adequate combination could potentially lead to interactions between vitamins that establish beneficial effects on bone development and calcium homeostasis in gilthead seabream juveniles.

## Figures and Tables

**Figure 1 animals-14-02808-f001:**
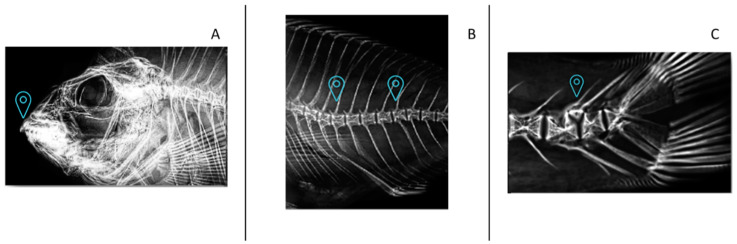
Main anomalies found in gilthead seabream juveniles fed with different levels of dietary vitamin D_3_ and vitamin K_3_ by X-ray analysis (from left to right, landmark indicates: (**A**) anomalous maxillary and/or pre-maxillary, (**B**) abdominal lordosis, (**C**) abdominal kyphosis, and supernumerary bone).

**Figure 2 animals-14-02808-f002:**
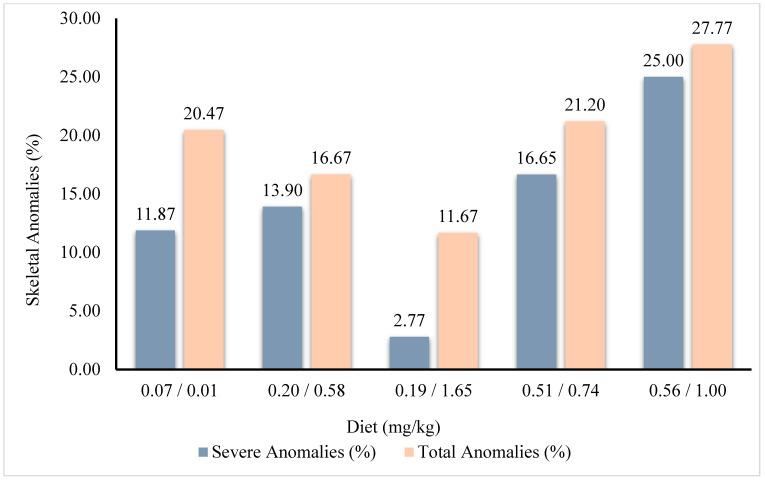
Percentage of incidence of severe and total skeletal anomalies in gilthead seabream juveniles fed with different levels of dietary vitamin D_3_ and vitamin K_3_.

**Figure 3 animals-14-02808-f003:**
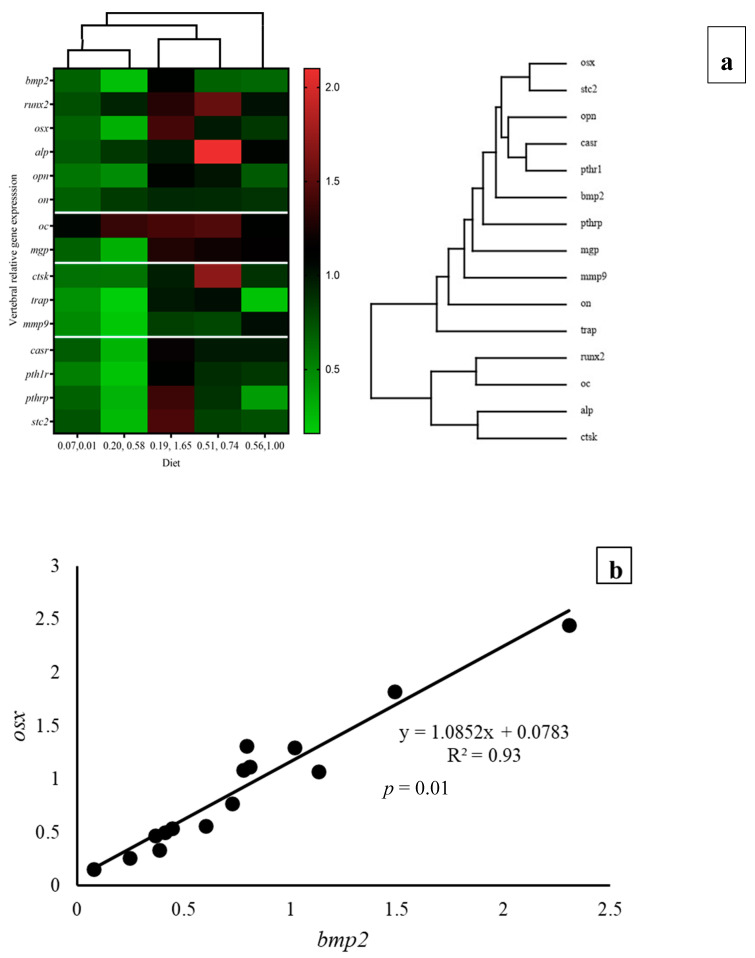
Heat map analysis of bone and calcium-related gene expression in vertebrae of gilthead seabream juveniles (**a**) and linear regression analysis between *bmp2* and *osx* gene expression (**b**) fed with different levels of dietary vitamin D_3_ and vitamin K_3_.

**Figure 4 animals-14-02808-f004:**
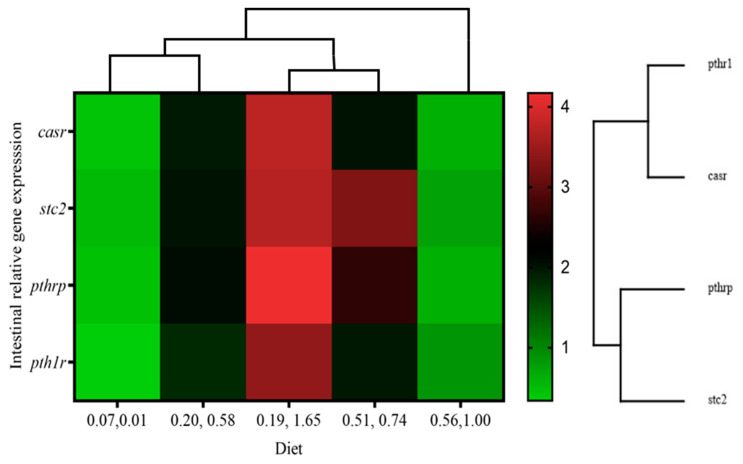
Heat map analysis of calcium-related gene expression in intestine of gilthead seabream juveniles fed with different levels of dietary vitamin D_3_ and vitamin K_3_.

**Figure 5 animals-14-02808-f005:**
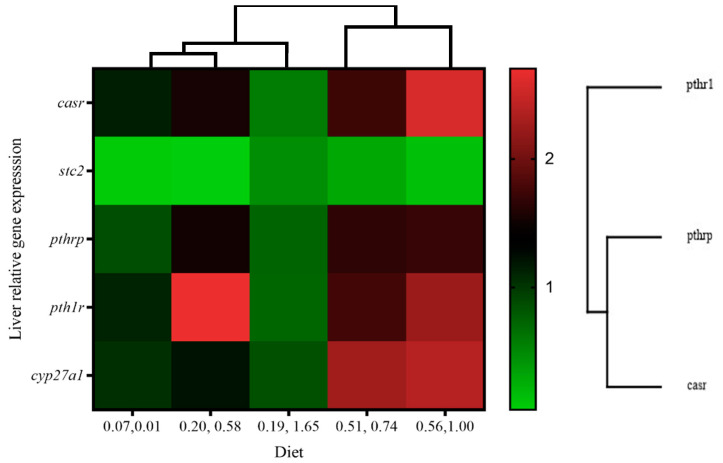
Heat map analysis of calcium-related gene expression in liver of gilthead seabream juveniles fed with different levels of dietary vitamin D_3_ and vitamin K_3_.

**Figure 6 animals-14-02808-f006:**
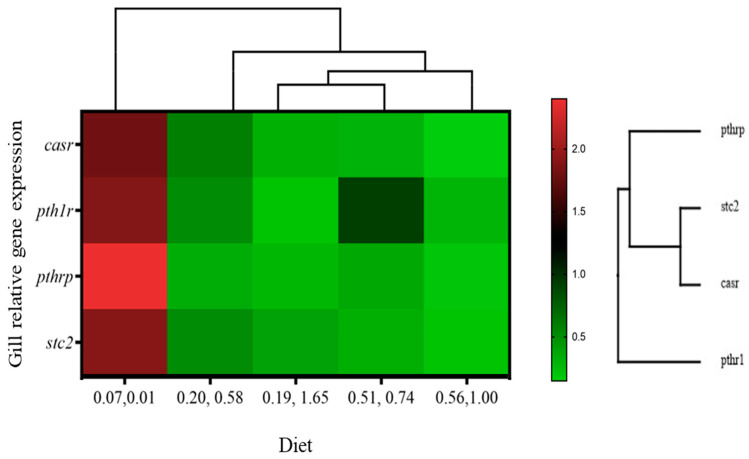
Heat map analysis of calcium-related gene expression in gills of gilthead seabream juveniles fed with different levels of dietary vitamin D_3_ and vitamin K_3_.

**Table 1 animals-14-02808-t001:** Ingredients and analyzed proximate composition of the experimental diets supplemented with different levels of vitamin D_3_ and vitamin K_3_.

Ingredients (%)	(0.07/0.01)	(0.20/0.58)	(0.19/1.65)	(0.51/0.74)	(0.56/1.00)
Fishmeal	10	10	10	10	10
Defatted squid meal	1	1	1	1	1
Squid meal	2	2	2	2	2
Casein	7.5	7.5	7.5	7.5	7.5
Soy protein concentrate	16	16	16	16	16
Wheat gluten meal	12.9	12.9	12.9	12.9	12.9
Corn gluten meal	12.9	12.9	12.9	12.9	12.9
Soybean meal	6.5	6.5	6.5	6.5	6.5
Wheat, whole	9	9	9	9	9
Rapeseed oil	3.6	3.6	3.6	3.6	3.6
Fish oil	6	6	6	6	6
Linseed oil	3.6	3.6	3.6	3.6	3.6
Soybean lecithin	2	2	2	2	2
L-Lysine	0.97	0.97	0.97	0.97	0.97
L-Methionine	0.27	0.27	0.27	0.27	0.27
L-Threonine	0.16	0.16	0.16	0.16	0.16
CaHPO_4_.2H_2_O	1	1	1	1	1
Carboxymethyl cellulose	0.6	0.6	0.6	0.6	0.6
Mineral premix *	2	2	2	2	2
Vitamin premix **	2	2	2	2	2
Vitamin D_3_ (mg/kg)	0	0.04	0.04	0.5	0.5
Vitamin K_3_ (mg/kg)	0	6	12	6	12
Diet proximate composition (%)					
Moisture	6.42 ± 0.13	6.28 ± 0.03	6.29 ± 0.04	6.41 ± 0.08	7.40 ± 0.05
Crude protein	44.81 ± 0.14	44.32 ± 0.37	44.60 ± 0.36	44.51 ± 0.02	44.34 ± 0.07
Crude lipid	21.64 ± 0.59	23.33 ± 0.46	22.74 ± 0.16	23.03 ± 1.15	21.57 ± 0.33
Ash	6.46 ± 0.08	6.20 ± 0.01	5.96 ± 0.24	6.10 ± 0.09	6.04 ± 0.02
Analyzed dietary vitamins					
Vitamin D_3_ (mg/kg)	0.07 ± 0.03	0.20 ± 0.04	0.19 ± 0.04	0.51 ± 0.10	0.56 ± 0.05
Vitamin K_3_ (mg/kg)	0.01 ± 0.00	0.58 ± 0.04	1.65 ± 0.05	0.74 ± 0.04	1.00 ± 0.06
Supplemented dietary vitamins					
Vitamin D_3_ (mg/kg)	0	0.04	0.04	0.5	0.5
Vitamin K_3_ (mg/kg)	0	6	12	6	12

* Supplied the following mineral (g/kg mix): calcium hydrogen phosphate dihydrate ((H_2_PO)_2_Ca)—1.605, calcium carbonate (CaCO_3_)—4, Iron(II) sulfate heptahydrate (FeSO_4_7H_2_O)—1.5, magnesium sulfate heptahydrate (MgSO_4_7H_2_O)—1.605, dipotassium hydrogen phosphate (K_2_HPO_4_)—2.8, disodium hydrogen phosphate hydrate (Na_2_PO_4_H_2_O)—1, aluminium sulfate hexahydrate (Al(SO_4_)_3_6H_2_O)—0.02, zinc sulfate pentahydrate (ZnSO_4_5H_2_O)—0.12, copper(II) sulfate pentahydrate (CuSO_4_5H_2_O)—0.12, potassium iodide (KI)—0.02, manganese(II) sulfate monohydrate (MnSO_4_H_2_O)—0.08, cobalt(II) sulfate heptahydrate (CoSO_4_7H_2_O)—0.08, and selenium (Se)—0.0003. ** Supplied the following vitamin (g/kg mix): vitamin B1 (thiamine)—0.06, vitamin B2 (riboflavin)—0.08, vitamin B6 (pyridoxine)—0.04, vitamin B5 (calcium pantothenate)—0.1169, vitamin B3 (nicotinic acid)—0.2, biotin (vitamin H)—0.001, vitamin B9 (Folic Acid)—0.04, vitamin B12 (cyanocobalamin)—0.0009, choline—2.7, myo-inositol—2, vitamin C (ascorbic acid)—5, vitamin E (α-tocopherol)—0.25, vitamin A (retinol acetate)—0.025, and ethoxyquin—0.1.

**Table 2 animals-14-02808-t002:** Genes and primer sequences used in the gene expression.

Gene	Forward Primer	Reverse Primer	Annealing Temperature T_m_	Accession Number
House Keeping Gene				
*β-actin2* (Beta actin 2)	TCTGTCTGGATCGGAGGCTC	AAGCATTTGCGGTGGACG	58.1	X89920
*ef1α* (Elongation factor 1 alpha)	CTTCAACGCTCAGGTCATCAT	GCACAGCGAAACGACCAAGGGGA	60	AF184170
*rpl27* (Ribosomal protein L27)	AAGAGGAACACAACTCACTGCCCCAC	GCTTGCCTTTGCCCAGAACTTTGTAG	68	AY188520
Bone Biomarker				
*bmp2* (Bone morphogenetic protein 2)	GTGGCTTCCATCGTATCAACATTTT	GCTCCCCGCCATGAGT	60	JF261172.1
*runx2* (Runt-related transcription factor 2)	GCCTGTCGCCTTTAAGGTGGTTGC	TCGTCGTTGCCCGCCATAGCTG	61	AJ619023
(*osx)* Osterix	CAGTCAGGGATTCAGCAACA	GGTGAAGGAGCCAGTGTAGG	60	ERR22591_isotig06993
*alp* (Alkaline phosphatase)	AGAACG CCCTGACGC TGCAA	TTCAGTATACGAGCA GCCGTCAC	61	AY266359
*opn* (Osteopontin)	AAGATGGCCTACGACATGACAGAC	CCTGAAGAGCCTTGTACACCTGC	61	AY651247
*on* (Osteonectin)	AAAATGATCGAGCCCTGCATGGAC	TACAGAGTCACCAGGACGTT	61	AY239014
*oc* (osteocalcin)	AGCCCAAAGCACGTAAGCAAGCTA	TTTCATCACGCTACTCTACGGGTT	58.1	AF048703
Calcium Regulating Gene				
*vdrβ* (Vitamin D receptor β)	TGACGACTCCTACTCTGACT	CTGCTGCCCTGCTCTTGGTA	62.3	XM_030420365.1
*cyp27a1* (Cytochrome P450 family 27 subfamily A member 1)	TGGCTCTACAAGTTTGGCTTTGA	TGAACCGCAGCG TGTCTTT	60	AM885865
*stc2* (Stanniocalcin 2)	CTGGAGCAAGTAGTGGGAT	CCTGTAGCCCTCGTATCTCG	62	XM_030395344
*casr* (Calcium sensing receptor)	GCTTCTCCAGCTCGCTCATC	AGGCGGGCTGGCGTAA	60	AJ289717
*pthrp* (Parathyroid hormone related protein)	GAGGCAAATGAA TGGAACAG	TGGCCAGCTCAA AACTTGT	60	AF197904
*pth1r* (Parathyroid hormone 1 receptor)	GAACCTGCCCGG CTACGTGAAG	GCTCCTGTCCCG ACGAGGGTAT	60	AJ619024
*mgp* (Matrix gla protein)	CGCCCGAAATACACCTCAGA	GACGGACGGATACTAGGAGTCTA	60	AY065652
*mmp9* (Matrix metallopeptidase 9)	ATTCAGAAGGTGGAGGGAGCG	CATTGGGGACACCACCGAAGA	60	AM905938
*ctsk* (Cathepsin K)	AGCGAGCAGAACCTGGTGGAC	GCAGAGTTGTAGTTGGGGTCGTAG	60	DQ875329
*trap* (Tartrate-resistant acid phosphatase)	CTTAATCGTTGCCATCCCTGTG	CTCCCATCTGCTCTGCTACTTTG	60	FM147928

**Table 3 animals-14-02808-t003:** Growth performance of gilthead seabream juveniles fed with different levels of dietary vitamin D_3_ and vitamin K_3_.

Growth Parameters	(0.07/0.01)	(0.20/0.58)	(0.19/1.65)	(0.51/0.74)	(0.56/1.00)	Two-Way ANOVA (*p* Value)		
						VD	VK	VD × VK
FBW (g/fish)	148.36 ± 6.34	147.99 ± 2.26	150.54 ± 3.45	141.64 ± 9.35	151.61 ± 5.78	0.99	0.23	0.72
WG (%)	103.32 ± 8.23	103.64 ± 3.72	108.28 ± 4.71	102.66 ± 0.86	109.13 ± 9.28	0.99	0.20	0.82
SGR (%)	0.68 ± 0.04	0.68 ± 0.02	0.70 ± 0.04	0.68 ± 0.01	0.70 ± 0.04	0.97	0.26	0.90
FCR	1.55 ± 0.08	1.56 ± 0.06	1.51 ± 0.05	1.54 ± 0.04	1.49 ± 0.09	0.64	0.26	1.00
PER	1.05 ± 0.05	1.05 ± 0.04	1.07 ± 0.04	1.06 ± 0.04	1.10 ± 0.06	0.60	0.27	0.64
FI (g/fish)	159.97 ± 5.29	161.47 ± 2.20	163.62 ± 3.81	154.85 ± 6.13	161.92 ± 4.55	0.34	0.27	0.74
CF	1.69 ± 0.01	1.71 ± 0.02	1.69 ± 0.04	1.69 ± 0.03	1.71 ± 0.04	0.80	0.80	0.35

**Table 4 animals-14-02808-t004:** Bone calcium and phosphorus content in gilthead seabream juveniles fed with different levels of dietary vitamin D_3_ and vitamin K_3_.

Serum Calcitriol and Bone Mineral Composition	(0.07/0.01)	(0.20/0.58)	(0.19/1.65)	(0.51/0.74)	(0.56/1.00)	Two-Way ANOVA *(p* Value)		
						VD	VK	VD × VK
Serum calcitriol (pg/mL)	5.26 ± 0.41	5.71 ± 0.07	5.61 ± 0.16	4.78 ± 0.76	5.16 ± 0.22	0.05	0.64	0.44
Bone calcium (mg/g)	113.33 ± 5.77	109.00 ± 11.53	106.33 ± 11.85	113.33 ± 5.77	110.00 ± 0.00	0.42	0.54	0.95
Bone phosphorus (mg/g)	57.33 ± 3.06	58.00 ± 6.25	55.67 ± 5.51	57.67 ± 2.52	57.67 ± 1.53	0.74	0.64	0.64

(number of fish sampled = 3 per tank).

**Table 5 animals-14-02808-t005:** Frequency of skeletal anomalies in gilthead seabream juveniles fed with different levels of dietary vitamin D_3_ and vitamin K_3_.

Skeletal Anomaly Frequency (%)	(0.07/0.01)	(0.20/0.58)	(0.19/1.65)	(0.51/0.74)	(0.56/1.00)	Kruskal-Wallis Test
Haemal Lordosis	0.00 ± 0.00	2.77 ± 4.79	0.00 ± 0.00	0.00 ± 0.00	8.33 ± 14.43	0.58
Haemal vertebral fusion	0.00 ± 0.00	0.00 ± 0.00	0.00 ± 0.00	0.00 ± 0.00	2.77 ± 4.79	0.45
Caudal vertebral anomaly	3.03 ± 5.25	0.00 ± 0.00	0.00 ± 0.00	0.00 ± 0.00	5.57 ± 9.64	0.58
Anomalous maxillary and/or pre-maxillary	6.07 ± 10.51	11.13 ± 9.64	2.77 ± 4.79	8.35 ± 11.81	8.33 ± 8.35	0.86
Anomalous dentary	2.77 ± 4.79	0.00 ± 0.00	0.00 ± 0.00	8.35 ± 11.81	0.00 ± 0.00	0.36
Caudal fin supernumerary bone	8.60 ± 8.36	2.77 ± 4.79	8.90 ± 8.40	4.55 ± 6.43	2.77 ± 4.79	0.51
Cranium	8.83 ± 9.11	11.13 ± 9.64	2.77 ± 4.79	16.65 ± 23.55	8.33 ± 8.35	0.80
Haemal vertebrae	0.00 ± 0.00	2.77 ± 4.79	0.00 ± 0.00	0.00 ± 0.00	11.10 ± 19.23	0.58
Caudal vertebrae	3.03 ± 5.25	0.00 ± 0.00	0.00 ± 0.00	0.00 ± 0.00	5.57 ± 9.64	0.58
Total severe anomaly	11.87 ± 14.00	13.90 ± 12.73	2.77 ± 4.79	16.65 ± 23.55	25.00 ± 22.05	0.64
Total anomaly	20.47 ± 18.62	16.67 ± 16.65	11.67 ± 12.58	21.20 ± 17.11	27.77 ± 25.46	0.87

(number of fish sampled = 10 per tank).

**Table 6 animals-14-02808-t006:** Morphometric traits in gilthead seabream juveniles fed with different levels of dietary vitamin D_3_ and vitamin K_3_.

Fish Morphometric Traits	(0.07/0.01)	(0.20/0.58)	(0.19/1.65)	(0.51/0.74)	(0.56/1.00)	Two-Way ANOVA (*p* Value)		
						VD	VK	VD × VK
Total lateral area (cm^2^)	98.47 ± 4.70	97.13 ± 2.44	96.22 ± 1.46	95.47 ± 5.08	97.45 ± 2.60	0.92	0.80	0.50
Total lateral length (cm)	20.48 ± 0.18	20.32 ± 0.30	20.17 ± 0.11	20.08 ± 0.57	20.20 ± 0.31	0.58	0.95	0.51
Fish maximum height (cm)	7.15 ± 0.13	7.16 ± 0.17	7.06 ± 0.02	7.05 ± 0.26	7.12 ± 0.10	0.79	0.84	0.35
Caudal peduncle height (cm)	2.34 ± 0.48	2.34 ± 0.34	2.13 ± 0.04	2.14 ± 0.17	2.21 ± 0.11	0.71	0.65	0.41
Fork length (cm)	20.29 ± 0.19	20.09 ± 0.29	19.91 ± 0.12	19.85 ± 0.58	20.02 ± 0.30	0.74	0.98	0.39
Head height (cm)	5.96 ± 0.09	6.11 ± 0.28	6.20 ± 0.12	6.00 ± 0.14	6.20 ± 0.23	0.59	0.20	0.61
Fillet area (cm^2^)	67.65 ± 2.55	65.68 ± 1.18	63.91 ± 0.66	64.21 ± 4.28	64.72 ± 1.02	0.81	0.65	0.42
Fillet area percentage (%)	0.69 ± 0.02	0.68 ± 0.01	0.66 ± 0.02	0.67 ± 0.01	0.66 ± 0.01	0.81	0.11	0.81
Standard length (cm)	17.61 ± 0.27	17.42 ± 0.22	17.34 ± 0.06	17.27 ± 0.42	17.37 ± 0.30	0.73	0.95	0.60
Fillet maximum length (cm)	11.60 ± 0.21	11.32 ± 0.08	11.08 ± 0.11	11.20 ± 0.38	11.15 ± 0.12	0.85	0.27	0.45
Fish eccentricity	0.89 ± 0.00	0.89 ± 0.00	0.89 ± 0.00	0.89 ± 0.01	0.89 ± 0.00	0.29	0.29	0.29
Head eccentricity	0.65 ± 0.01	0.66 ± 0.04	0.69 ± 0.02	0.66 ± 0.03	0.68 ± 0.02	0.74	0.12	0.91
Tail-excluded length (cm)	15.51 ± 0.21	15.40 ± 0.28	15.31 ± 0.03	15.23 ± 0.38	15.35 ± 0.21	0.66	0.90	0.48
Equidistant fish height A	2.89 ± 0.28	2.86 ± 0.23	2.86 ± 0.12	2.90 ± 0.09	2.95 ± 0.02	0.52	0.77	0.80
Equidistant fish height B	4.26 ± 0.27	4.25 ± 0.02	4.27 ± 0.13	4.24 ± 0.17	4.35 ± 0.12	0.71	0.52	0.68
Equidistant fish height C	6.40 ± 0.16	6.36 ± 0.06	6.34 ± 0.08	6.26 ± 0.22	6.34 ± 0.11	0.55	0.72	0.52
Equidistant fish height D	7.10 ± 0.14	7.07 ± 0.11	7.02 ± 0.03	7.00 ± 0.27	7.08 ± 0.10	0.93	0.90	0.46
Equidistant fish height E	5.68 ± 0.09	5.67 ± 0.06	5.59 ± 0.03	5.58 ± 0.20	5.62 ± 0.07	0.62	0.73	0.31

(number of fish sampled = 10 per tank).

**Table 7 animals-14-02808-t007:** Relative gene expression of vertebral bone biomarkers and calcium regulators in gilthead seabream juveniles fed with different levels of dietary vitamin D_3_ and vitamin K_3_.

Relative Gene Expression—Vertebra						Two-Way ANOVA (*p* Value)		
Gene	(0.07/0.01)	(0.20/0.58)	(0.19/1.65)	(0.51/0.74)	(0.56/1.00)	VD	VK	VD × VK
Bone Biomarkers (Osteoblast-Related Genes)								
*bmp2*	0.67 ± 0.09 ^ab^	0.23 ± 0.15 ^a^	1.11 ± 0.44 ^b^	0.67 ± 0.22 ^ab^	0.64 ± 0.42 ^ab^	0.94	0.00	0.00
*runx2*	0.75 ± 0.25	0.94 ± 0.29	1.30 ± 0.91	1.54 ± 0.64	1.02 ± 0.70	0.57	0.76	0.12
*osx*	0.67 ± 0.19 ^a^	0.30 ± 0.16 ^a^	1.41 ± 0.39 ^b^	0.98 ± 0.56 ^ab^	0.86 ± 0.41 ^ab^	0.69	0.01	0.00
*alp*	0.70 ± 0.22 ^a^	0.86 ± 0.42 ^a^	0.98 ± 0.43 ^a^	2.10 ± 0.42 ^b^	1.08 ± 0.56 ^a^	0.00	0.02	0.00
*opn*	0.58 ± 0.08 ^a^	0.47 ± 0.25 ^a^	1.08 ± 0.30 ^b^	1.00 ± 0.54 ^ab^	0.70 ± 0.52 ^a^	0.66	0.34	0.01
*on*	0.68 ± 0.26	0.84 ± 0.33	0.92 ± 0.35	0.91 ± 0.27	0.87 ± 0.29	0.92	0.87	0.61
*oc*	1.05 ± 0.33	1.36 ± 1.18	1.42 ± 0.49	1.46 ± 0.50	1.10 ± 0.57	0.74	0.64	0.51
Bone Biomarkers (Osteoclast-Related Genes)								
*mgp*	0.67 ± 0.09 ^ab^	0.30 ± 0.11 ^a^	1.28 ± 0.34 ^c^	1.22 ± 0.14 ^c^	1.14 ± 0.38 ^bc^	0.00	0.00	0.00
*ctsk*	0.60 ± 0.10 ^a^	0.59 ± 0.12 ^a^	0.96 ± 0.53 ^a^	1.70 ± 0.92 ^b^	0.88 ± 0.34 ^a^	0.06	0.39	0.03
*trap*	0.44 ± 0.06 ^a^	0.16 ± 0.04 ^a^	0.99 ± 0.64 ^b^	1.03 ± 0.83 ^b^	0.21 ± 0.13 ^a^	0.84	0.98	0.00
*mmp9*	0.48 ± 0.10	0.19 ± 0.11	0.82 ± 0.36	0.78 ± 0.50	1.03 ± 0.85	0.07	0.05	0.37
Calcium Regulators								
*casr*	0.69 ± 0.18 ^ab^	0.28 ± 0.15 ^a^	1.18 ± 0.28 ^c^	0.98 ± 0.12 ^bc^	0.98 ± 0.36 ^bc^	0.03	0.00	0.00
*pth1r*	0.53 ± 0.12 ^ab^	0.21 ± 0.09 ^a^	1.10 ± 0.35 ^b^	0.90 ± 0.23 ^b^	0.86 ± 0.50 ^b^	0.10	0.01	0.00
*pthrp*	0.67 ± 0.25 ^ab^	0.29 ± 0.21 ^a^	1.39 ± 0.36 ^c^	0.88 ± 0.34 ^bc^	0.39 ± 0.07 ^ab^	0.10	0.02	0.00
*stc2*	0.73 ± 0.13 ^a^	0.25 ± 0.10 ^a^	1.44 ± 0.64 ^b^	0.81 ± 0.35 ^ab^	0.75 ± 0.30 ^a^	0.68	0.00	0.00

Different letters (a, b, c) in the row denote significant differences between groups fed different diets (mean ± SD, n = 3, *p* < 0.05). (number of fish sampled = 3 per tank).

**Table 8 animals-14-02808-t008:** Relative gene expression of intestinal calcium regulators in gilthead seabream juveniles fed with different levels of dietary vitamin D_3_ and vitamin K_3_.

Relative Gene Expression—Intestine						Two-Way ANOVA (*p* Value)		
Gene	(0.07/0.01)	(0.20/0.58)	(0.19/1.65)	(0.51/0.74)	(0.56/1.00)	VD	VK	VD × VK
*casr*	0.43 ± 0.11 ^a^	1.96 ± 0.85 ^b^	3.75 ± 1.31 ^c^	2.01 ± 0.41 ^b^	0.62 ± 0.1 ^ab^	0.01	0.53	0.00
*pth1r*	0.34 ± 0.04 ^a^	1.83 ± 1.04 ^b^	3.42 ± 0.65 ^c^	1.98 ± 0.80 ^b^	0.87 ± 0.36 ^ab^	0.16	0.80	0.01
*pthrp*	0.46 ± 0.09 ^a^	2.07 ± 1.43 ^ab^	4.17 ± 1.13 ^b^	2.66 ± 1.48 ^ab^	0.62 ± 0.15 ^a^	0.30	0.26	0.01
*stc2*	0.52 ± 0.25 ^a^	2.02 ± 1.22 ^ab^	3.71 ± 1.42 ^b^	3.28 ± 1.09 ^b^	0.76 ± 0.27 ^a^	0.44	0.23	0.00
*pxr*	0.50 ± 0.32	1.95 ± 0.69	2.30 ± 1.42	1.92 ± 0.20	1.97 ± 3.05	0.85	0.84	0.88

Different letters (a, b, c) in the row denote significant differences between groups fed different diets (mean ± SD, n = 3, *p* < 0.05). (number of fish sampled = 3 per tank).

**Table 9 animals-14-02808-t009:** Relative gene expression of kidney calcium regulators in gilthead seabream juveniles fed with different levels of dietary vitamin D_3_ and vitamin K_3_.

Relative Gene Expression—Kidney						Two-Way ANOVA (*p* Value)		
Gene	(0.07/0.01)	(0.20/0.58)	(0.19/1.65)	(0.51/0.74)	(0.56/1.00)	VD	VK	VD × VK
*vdrβ*	0.72 ± 0.29 ^ab^	0.51 ± 0.34 ^a^	0.42 ± 0.20 ^a^	0.87 ± 0.41 ^b^	0.94 ± 0.37 ^c^	0.02	0.79	0.34
*stc2*	0.55 ± 0.14 ^ab^	0.22 ± 0.13 ^a^	3.54 ± 0.37 ^d^	1.07 ± 0.22 ^b^	1.96 ± 0.51 ^c^	0.14	0.00	0.00

Different letters (a, b, c, d) in the row denote significant differences between groups fed different diets (mean ± SD, n = 3, *p* < 0.05). (number of fish sampled = 3 per tank).

**Table 10 animals-14-02808-t010:** Relative gene expression of liver calcium regulators in gilthead seabream juveniles fed with different levels of dietary vitamin D_3_ and vitamin K_3_.

Relative Gene Expression—Liver						Two-Way ANOVA (*p* Value)		
Gene	(0.07/0.01)	(0.20/0.58)	(0.19/1.65)	(0.51/0.74)	(0.56/1.00)	VD	VK	VD × VK
*casr*	1.14 ± 0.73	1.54 ± 0.81	0.58 ± 0.27	1.73 ± 1.39	2.57 ± 1.65	0.35	0.23	0.55
*pth1r*	1.12 ± 0.67	2.70 ± 1.59	0.71 ± 0.12	1.75 ± 1.04	2.24 ± 0.83	0.48	0.19	0.34
*pthrp*	0.86 ± 0.19	1.52 ± 0.76	0.72 ± 0.31	1.65 ±1.20	1.70 ± 0.31	0.21	0.40	0.34
*stc2*	0.06 ± 0.05	0.05 ± 0.03	0.46 ±0.49	0.30 ± 0.39	0.13 ± 0.08	0.26	0.57	0.74
*cyp27a1*	1.04 ± 0.33	1.21 ± 0.49	0.85 ± 0.35	2.27 ± 1.64	2.38 ± 0.92	0.05	0.73	0.77

(number of fish sampled = 3 per tank).

**Table 11 animals-14-02808-t011:** Relative gene expression of gill calcium regulators in gilthead seabream juveniles fed with different levels of dietary vitamin D_3_ and vitamin K_3_.

Relative Gene Expression—Gill						Two-Way ANOVA (*p* Value)		
Gene	(0.07/0.01)	(0.20/0.58)	(0.19/1.65)	(0.51/0.74)	(0.56/1.00)	VD	VK	VD × VK
*casr*	1.80 ± 1.01 ^b^	0.58 ± 0.16 ^a^	0.31 ± 0.20 ^a^	0.30 ± 0.36 ^a^	0.15 ± 0.07 ^a^	0.24	0.03	0.65
*pth1r*	1.89 ± 0.71 ^b^	0.51 ± 0.33 ^a^	0.21 ± 0.08 ^a^	0.92 ± 0.64 ^a^	0.28 ± 0.13 ^a^	0.74	0.01	0.94
*pthrp*	2.40 ± 1.00	0.33 ± 0.06	0.27 ± 0.16	0.36 ± 0.33	0.19 ± 0.09	0.88	0.11	0.51
*stc2*	1.91 ± 1.05 ^b^	0.51 ± 0.25 ^a^	0.39 ± 0.21 ^a^	0.32 ± 0.20 ^a^	0.21 ± 0.08 ^a^	0.27	0.03	0.58

Different letters (a, b) in the row denote significant differences between groups fed different diets (mean ± SD, n = 3, *p* < 0.05). (number of fish sampled = 3 per tank).

## Data Availability

The raw data supporting the conclusions of this article will be made available by the authors upon request.

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
