# Peer review of "Effects of the Interaction between Dietary Vitamin D3 and Vitamin K3 on Growth, Skeletal Anomalies, and Expression of Bone and Calcium Metabolism-Related Genes in Juvenile Gilthead Seabream (Sparus aurata)"

_animals, 2024, doi:10.3390/ani14192808_

Round 1

Reviewer 1 Report

Comments and Suggestions for Authors

Comments overall

This study contributes much needed knowledge of vitamins for a commercially important fish species. To date, fundamental requirements for both vitamin D and K in sea bream and other species are still not certain. An excellent aspect of this study is the consideration given to vitamin interactions. The potential interactions between vitamins D & K are certainly of interest. 

The study certainly provides some insights into fundamental D and K requirements and importantly also under the scenario of co-supplementation. A lack of effects were found in zootechnical performance, vitamin D hormone, body morphometrics and it appears also bone anomalies. Bone anomalies seemed high in general in the study, especially given the fish were graded against gross skeletal anomalies before experimental feeding. Did the basal diet truly met all requirements and may have the genetic fitness of this stock been suboptimal? Whilst the study did measure Ca and P in the vertebrae, Ca and P in the plasma could have re-enforced the findings, particularly for example of transcriptional gill responses. Long-term plasma Ca and P can, however, be quite stable across a range of dietary vitamin D concentrations. The only evidence of any significant changes occurring are in the gene expression. It is worth noting that some important genes relating to vitamin D metabolism are missing from the study.

The study certainly has merits, but should be re-analyzed and re-interpreted according to statistical output. Could a multivariate be used, in particular linking the bone anomalies to gene expression? A major challenge in assessing vitamin D requirements and optimal status is the lack of markers available and the uncertainty of the requirements for different physiological processes. Growth is still mostly likely amongst the more reliable markers of minimum requirements. How closely growth and calcitriol match in this regard is still not clear for fish. A hierarchy of requirements likely exists, which may depend on the species in question. The lack of effects seen in several parameters is valuable data worthy of publication. For example, requirements for immunity and the metabolism of other nutrients may well be different than for bone health, growth, or Ca handling.

A significant part of the study covers and interprets findings from gene expression data. These data are certainly of interest, but other aspects of the study deserve equal attention, in particular the lack of effects on growth, morphometrics (and bone anomalies) and of calcitriol. Whilst not positive, these are interesting and valuable findings.

Other specific comments

The authors did a great job to ensure that the feeds would retain the levels of vitamin after processing. Whilst scientifically sound, the caveat to this approach is that in commercial feeds, nominal to recovered vitamin K levels will be rather different. The reader should be made aware of this.

The formulations used in the current study differ from ‘average’ commercial feeds. Consequently, can the authors give the reader an indication as to how the findings of this study may relate to commercial realities, i.e. typical basal levels of these vitamins.

The study should better indicate the rational behind the levels used and the ratios between vitamin D and K, especially in the abstract.

The most important stage for the assessment of bone deformities is surely at the larval stages. At the size tested in the current study, vitamin effects on deformities will be much reduced. Thus is it surprising that results were not significantly different? This could be mentioned in the study.

Plasma calcitriol relates to the endocrine functions of vitamin D. Some of the effects seen in the present study may be intracrine/paracrine. For example, 1 alpha-hydroxylase activity in osteoclasts may metabolise calcitriol in this species. This may explain some of the disconnect between transcriptional responses and other measures in the study. The introduction indeed mentions ‘classical & non-classical vitamin D metabolism’. The number of samples measured for calcitriol is quite low in the present study. Calcitriol is the most variable of the vitamin D metabolites (in salmonids). How robust is the ANOVA, is the power above 80%, there appears to be a risk of a type I error here.  

The discussion should start with an interpretation of growth performance etc. A large part of the discussion is given to interpretation of gene expression data. The transcriptional data should be interpreted with caution. The single correlation in figure 3 is a nice insight, are there other correlation in the genes set? Transcription is only one method by which protein activity and physiological processes are controlled. Post-translation modifications, protein location within cells, non-genomic regulation etc. all play their roles. In future studies and where feasible, transcriptional data should be backed up with some biochemical markers, for example, alp expression with alp enzyme activity. In a similar respect, a limitation of the study is the lack of plasma Ca and P data that could have complemented data on calcium regulating genes.

Line 1. Title: Does interplay between vitamin D and K really describe the findings of the study? ‘Calcium regulating gene (s)’ does not well describe the panel of genes tested.

Line 14. This sentence on a lack of R&D could be dropped. In its place a sentence on the why D3 and K may interact?

Line 16. Use of ‘explore’ – rather state succinct hypotheses

Line 18. Using the measured values is transparent, but it is difficult to quickly understand the experimental approach being used.  Perhaps use nominal values or include the ratio between D3 and K3? Another descriptor may make it easier for the reader to appreciate the experimental approach.

Lince 27. ‘gene expression’

Line 42. There is evidence that some fish species can synthesize (some) D3, see Pierens & Fraser 2015 (trout).

Line 43. The adipose tissue, and muscle due to its size, also store significant quantities of vitamin D. In its storage form vitamin D can be esterified.

Line 111. ‘caught’. This sentence needs attention - researchers in biomedical fields…

Line 133. Effects on bone health particularly relevant in larval fish as this tissue is forming and developing.

Line 135. But the current study used a rather different feed formulation and food processing conditions than occurs commercially. This should be clear to the reader, which may be a reading in the interest of optimizing commercial feeds.

Line 293. Due to the number of genes tested, a correction for multiplicity testing should be performed.

Line 329. There is almost a difference which should be addressed in the results text. This group has the highest variability. Is there an outlier in this data, has it been subjected to a Grubb’s test? Quite a different approach has been taken for the statistical interpretation of calcitriol result compared to the bone anomalies…

Line 341. The authors talk of ‘tendencies’ in skeletal deformities, but the bottom-line is that the p values are insignificant by quite a margin. As analyzed, this data set be interpreted as no effect observed, which changes somewhat the outcomes of the study. How about using a multi-factorial statistics, there could be something in this data, but not with these tests & p values – they are insignificant. Do any of these anomalies correlate to the gene expression data? This is a major issue with the study.

Line 544. Inversely, perhaps the VDR and stc did regulate calcitriol making sure it remained stable?

Figure 2. Should show standard error bars.

Comments on the Quality of English Language

The manuscript is quite well written but should be given another check for minor typos and grammatical errors.

Reviewer 2 Report

Comments and Suggestions for Authors

This study investigated the interactive effects of dietary VD and VK levels on the growth performance, bone biomarkers, and the transcriptions of genes involved in calcium regulating Gene in juvenile gilthead seabream. Generally, the topic is quite interesting, and can attract broad interest. The manuscript was well organized, clearly written and easy to understand with systematic data presented. However, several concerns should be addressed to improve the quality of the manuscript. Please refer to the following specific comments.

Major ones:

1.      This study was conducted in a practical perspective. However, the feed formulation is a little bit puzzling concerning the use of casein. Is this ingredient commonly adopted in the commercial feed of gilthead seabream?

2.      In this study, a total of three housekeeping genes were adopted to normalize the relative gene expression in different tissues. However, no corresponding explanation was provided. For example, β-actin 2 was used to normalize the expressions of which genes, or the transcriptions of genes in which tissue?

3.      Hepatic VD and VK contents should be measured to reveal the effect of dietary manipulations on the tissue deposition of both vitamins.

4.      The data presented in Table 7 are repetitive to those in Figure 3A. This also holds true for the following tables and figures. Deletion of Figure 3 A and the following ones is recommended.

5.      The SD bars were missing in Figure 2.

6.      Table 7, the multiple comparisons analysis is missing for the data of opn.

7.      Please double check the Reference section to avoid careless mistakes. For example, the journal name of Comparative Biochemistry and Physiology Part A should be presented in the abbreviated form in reference 5; “Salmo Salar” should be italic in reference 62; etc.

8.      English in the text should be revised extensively since countless mistakes have been found. For example, the statements in lines 60-66; Line 466, “studies” should be “study”, and so on.

Minor ones:

1.      Please apply subscript formatting to the numbers in the various minerals in the footnote of Table 1, like CaCO3, FeSO47H2O, etc.

2.      The names of the genes presented in Table 2 should be double checked. For example, “Vdrβ” should be “vdrβ”; “Ribosomal Protein L27” should be “Ribosomal protein L27”; “Cytochrome P450 Family 27 Subfamily A Member 1” should be “Cytochrome P450 family 27 subfamily A member 1”; etc.

3.      Table 3, please provide the unit of FI.

4.      Please delete the P values in the Discussion and Conclusions sections.

Comments on the Quality of English Language

 Moderate editing of English language required.

Reviewer 3 Report

Comments and Suggestions for Authors

The study investigated the effects of various dietary combinations of Vitamin D3 and K3 on juvenile gilthead seabream. Results indicated that gene expression related to bone metabolism was influenced by the dietary combinations. The most successful dosage in terms of gene expression related to bone formation was the combination of 0.19 mg/kg of Vitamin D3 and 1.65 mg/kg of Vitamin K3, as it showed the highest levels of expression of the genes bmp2 and osx, which are important for osteogenesis.

Introduction: This is a well-developed section that effectively includes relevant literature and clearly identifies the current research gap.

Methods: This is a solid section with a good description of the diets and fish experiment. However, the evaluation of abnormalities should include a more detailed explanation.

Results: It is a good section

Discussion This is a well-constructed section that effectively compares the results with those of other studies and the current information available.

Round 2

Reviewer 1 Report

Comments and Suggestions for Authors

I am still of the opinion that due to a lack of significance the bone deformities are over interpreted.  The author's did not really address this. All other changes were good.

Author Response

The authors agree with the reviewer and have thus made changes in the text accordingly. Specifically, on the abstract and the discussion.

Other changes have also been made in order to adhere to the journal’s standards, including:
•    Change in the title
•    Inclusion of a new section (3.5) to mention the skeletal anomalies
•    Expression of  gene cyp27a1 in table 10
•    Changes in the format of several genes (to italics)
•    Correction of certain expression errors and minor punctuation errors
•    Correction of the vitamin D3 and K3 to D3 and K3
•    Inclusion of funding statement for Dr. David Dominguez
We hope these changes improve the manuscript and are worthy of your approval.

Reviewer 2 Report

Comments and Suggestions for Authors

The authors have made substantial efforts to revise the manuscript. All the comments were responded appropriately. It can be accepted for publication.

Author Response

Dear editors and reviewers, the authors acknowledge the effort made by you in order to improve the manuscript. In accordance with your inquiries, we have generated the attached document.